# Insights into CSF-1R Expression in the Tumor Microenvironment

**DOI:** 10.3390/biomedicines12102381

**Published:** 2024-10-18

**Authors:** Caterina Tomassetti, Gaia Insinga, Francesca Gimigliano, Andrea Morrione, Antonio Giordano, Emanuele Giurisato

**Affiliations:** 1Department of Biotechnology Chemistry and Pharmacy, University of Siena, 53100 Siena, Italy; c.tomassetti@student.unisi.it; 2Department of Medical Biotechnologies, University of Siena, 53100 Siena, Italy; giordano12@unisi.it; 3Department of Mental and Physical Health and Preventive Medicine, University of Campania “Luigi Vanvitelli”, 80138 Napoli, Italy; gaia.insinga@unicampania.it (G.I.); francesca.gimigliano@unicampania.it (F.G.); 4Sbarro Institute for Cancer Research and Molecular Medicine, Center for Biotechnology, Department of Biology, College of Science and Technology, Temple University, Philadelphia, PA 19122, USA; andrea.morrione@temple.edu

**Keywords:** CSF-1R, tumor microenvironment, CAFs, ECs, TAMs, MDSCs, cancer stem cells

## Abstract

The colony-stimulating factor 1 receptor (CSF-1R) plays a pivotal role in orchestrating cellular interactions within the tumor microenvironment (TME). Although the CSF-1R has been extensively studied in myeloid cells, the expression of this receptor and its emerging role in other cell types in the TME need to be further analyzed. This review explores the multifaceted functions of the CSF-1R across various TME cellular populations, including tumor-associated macrophages (TAMs), myeloid-derived suppressor cells (MDSCs), dendritic cells (DCs), cancer-associated fibroblasts (CAFs), endothelial cells (ECs), and cancer stem cells (CSCs). The activation of the CSF-1R by its ligands, colony-stimulating factor 1 (CSF-1) and Interleukin-34 (IL-34), regulates TAM polarization towards an immunosuppressive M2 phenotype, promoting tumor progression and immune evasion. Similarly, CSF-1R signaling influences MDSCs to exert immunosuppressive functions, hindering anti-tumor immunity. In DCs, the CSF-1R alters antigen-presenting capabilities, compromising immune surveillance against cancer cells. CSF-1R expression in CAFs and ECs regulates immune modulation, angiogenesis, and immune cell trafficking within the TME, fostering a pro-tumorigenic milieu. Notably, the CSF-1R in CSCs contributes to tumor aggressiveness and therapeutic resistance through interactions with TAMs and the modulation of stemness features. Understanding the diverse roles of the CSF-1R in the TME underscores its potential as a therapeutic target for cancer treatment, aiming at disrupting pro-tumorigenic cellular crosstalk and enhancing anti-tumor immune responses.

## 1. Introduction

The colony-stimulating factor 1 receptor (CSF-1R) is a tyrosine kinase receptor which plays a significant role in the regulation of the immune system, particularly in the development and function of macrophages. The CSF-1R is activated by its ligands, colony-stimulating factor 1 (CSF-1) and Interleukin-34 (IL-34), and these interactions are crucial for the survival, proliferation, and differentiation of mononuclear phagocytes [1]. The CSF-1R is a transmembrane receptor consisting of an extracellular ligand-binding region and an intracellular portion. Indeed, ligand binding leads to the dimerization of the receptor, which is present on the membrane in monomeric form, resulting in the internalization and autophosphorylation of the tyrosine kinase domains present in the intracellular portion. This is followed by the phosphorylation and cascade activation of various downstream signaling pathways, the major ones of which are ERK5-Akt, ERK1-2, PI3K-STAT1, PI3K-AKT, and PLCγ. All these pathways are critical for the survival, differentiation, and maturation of myeloid cells [1,2]. While the role of the CSF-1R in myeloid cells has been extensively reviewed in many articles, there is a lack of in-depth analysis on its role in individual components of the TME, particularly in non-myeloid stromal and immune cells. Thus, this review focuses on the expression and function of the CSF-1R in the various cellular components of the TME and analyzes the emerging role of CSF-1R/CSF-1R ligands in the crosstalk between infiltrating immune cells, stromal, and cancer cells.

### 1.1. CSF-1R Expression in Cancer

CSF-1R expression has been recently characterized in various neoplasia, where it correlates with worse prognosis, and a more immunosuppressive and aggressive tumor phenotype. As highlighted in a recent review, the CSF-1R is expressed at the cell membrane of cancer cells, where its activation mediates cancer progression through several signaling pathways, thereby promoting proliferation, survival, drug resistance, and the maintenance of cancer stemness [3]. Interestingly, the peculiar regulation of this receptor in cancer cells has been observed, as in fact, cancer cells sustain an autocrine regulatory mechanism whereby they produce its main ligand CSF-1, which in turn binds to the high number of receptors expressed on their surface, enhancing its activity in these cells. In addition, cancer cells possess mutations in the extracellular region of the receptor, leading to the constitutive expression of the CSF-1R and the sustained activation of its tyrosine kinase activity [4,5]. This results in a prolonged growth signal promoting cancer cell proliferation. Notably, this receptor is not uniformly distributed across all tumor cells but is expressed in a subset of cells exhibiting particularly aggressive behavior, suggesting its contribution to tumor malignancy [6,7]. In these cells, CSF-1R expression leads to increased proliferation, the expression of epithelial to mesenchymal transition (EMT) markers, enhanced invasive capacity, drug resistance, and the acquisition of stem-like features [8,9,10,11]. This has led to the consideration that anti-CSF-1R therapies could be beneficial not only in reducing the presence of M2-like tumor-associated macrophages (TAMs) but also in targeting cancer cells themselves. However, detecting the CSF-1R in cancer cells is challenging due to its low expression level when compared to myeloid cells, particularly in tumor tissues dominated by CSF-1R^+^ monocytes/macrophages. Thus, it is crucial to delve deeper into this area of research to address unresolved questions, such as the conditions under which the CSF-1R is expressed, its regulatory mechanisms of activation in different cancer types, and whether its expression is confined to a specific cell population or varies throughout the cell’s life cycle. Additionally, investigating whether CSF-1R regulation differs from other tumor cell receptors is essential. Thus, understanding the molecular mechanisms governing CSF-1R expression and function in cancer cells, and whether these mechanisms mirror those in immune cells, is fundamental for developing new CSF-1R-targeted cancer therapies.

### 1.2. CSF-1R in Inflammation-Induced Cancer

As previously reported, the CSF-1R is a typical myeloid receptor, but under inflammatory conditions it can undergo a different type of regulation, as happens in cancer cells and cells associated with the tumor microenvironment (TME). CSF-1R signaling is crucial for the chemotaxis, migration, and activation of many cells, especially immune cells, and significantly contributes to the progression of various inflammatory diseases and cancer, as reviewed by Xiang et al. [12]. Inhibiting the CSF-1R alters the TME to enhance the effectiveness of chemotherapy, radiation therapy, and immunotherapies, thereby reducing metastasis [13]. While this strategy has shown promising results, further research is needed to fully understand its potential and optimize its applications.

## 2. CSF-1R in Tumor-Associated Immune Cells

Within the TME, there is a consistent component of tumor-associated immune cells [14]. Since the CSF-1R has been mostly detected in cells of myeloid origin, its expression and role has often been described in monocytes/macrophages and myeloid-derived suppressor cells (MDSCs) [15]. However, within immune cells there are several myeloid-derived cells which are different from the monocytes/macrophage lineage, including megakaryocytes (MKs), dendritic cells (DCs), and granulocytes. Moreover, among lymphoid-derived cells, regulatory T cells (T_regs_) and natural killer (NK) cells are associated with the CSF-1/CSF-1R pathway in various aspects. Indeed, CSF-1R action has been supported in all these cell types either through its expression or its functional role (Figure 1). MKs, the polynucleated myeloid-derived hematopoietic cells that give rise to platelets, express the gene encoding *CSF-1R*. However, the precise function of the CSF-1R in these specific cells remains uncertain under both physiological conditions and within the TME [16].

In a study examining MK differentiation in mice fetal liver, CSF-1R expression was not detected during embryonic development, whereas it appeared during MK differentiation in adults. This finding suggests a possible involvement of the CSF-1R pathway in adult hematopoiesis, which may differ from its role in embryonic hematopoiesis [17]. Additionally, in the context of acute myeloid leukemia (AML), particularly the megakaryoblastic subtype, an in vitro study identified a genetic mutation leading to a RBM6-CSF-1R fusion. This mutation results in the cytoplasmic expression of the protein and the constitutive activation of CSF-1R kinase activity, thereby driving myeloproliferative disease [18]. Together, these observations highlight the necessity for further research to clarify both the functional implications of CSF-1R expression in MKs, and whether these cells could express CSF-1R proteins. Furthermore, *Csf-1r* mRNA expression was detected in granulocytes, but these cells likely do not express the protein [19,20]. Additionally, a study has shown that polymorphisms in the CSF-1R gene may be a marker of susceptibility to asthma, with a high expression of the CSF-1R in CD14^+^ monocytes and neutrophils in subjects with asthma compared to normal controls [21].

Based on previous evidence, current knowledge about the expression of this protein in MKs and neutrophils is limited, and the functional significance of the gene in these cells remains unclear, emphasizing that further studies are needed to clarify this. Additionally, although CSF-1R protein expression has not been detected in lymphoid-derived cells, this receptor has a significant role in CD4^+^ and CD8^+^ T_regs_. In fact, blocking the CSF-1/CSF-1R axis results in a decrease in CD4^+^ Foxp3^+^ T_regs_ correlated with a reduction in tumor growth [22] and enhances CD8^+^ T cell-mediated immunity in the sarcoma TME [23]. Likewise, studies have shown that T_regs_ express the CSF-1R [24,25] and these cells can produce IL-34, suggesting that IL-34 is the primary ligand for the CSF-1R in this population, functioning through autocrine mechanisms [25]. However, the specific signaling and role of this axis in T_regs_ remain unclear, necessitating further studies to elucidate its expression and function in these cells. Regarding other cells of lymphoid origin, it was demonstrated that blocking the CSF-1R led to an indirect decrease in the NK cell number, predominantly due to a shortage of myeloid cells responsible for supplying IL-15, the key survival factor for NK cells [26]. In summary, although some studies have indicated that the CSF-1R is expressed in various immune cells, including MKs, granulocytes, and T_regs_, its precise role remains unclear. Further studies are therefore essential to clarify the role of the CSF-1/CSF-1R axis in these cells of innate immunity which are part of the TME. As mentioned above, the CSF-1/CSF-1R axis is known to play a crucial role in MDSCs and TAMs, but it is likely important in DCs as well.

### 2.1. Tumor-Associated Macrophages

The precise regulation of macrophage phenotypes is pivotal in cancer. Macrophage differentiation relies on factors like CSF-1, colony-stimulating factor-2 (CSF-2 or GM-CSF), and various stimuli that define their functional polarization [27]. Macrophages residing within the TME, commonly known as TAMs, are abundantly present within tumoral areas in virtually every cancer type and represent a substantial proportion of TME cells [28]. Furthermore, their presence correlates with a worse prognosis in many solid tumors as in fact TAMs exert a significant impact on tumor initiation, progression, and metastasis and confer resistance to treatment [29,30,31,32]. Notably, this depends on their polarization, which is determined by local stimuli present in the TME, as in fact macrophages exhibit remarkable plasticity, particularly in the TME. The recruitment of TAMs to the tumoral area is dependent on immunomodulation exerted by tumor cells, which is mostly dependent on the chemokine (C-C motif) ligand 2 (CCL2), vascular endothelial growth factor (VEGF), platelet-derived growth factor (PDGF), and CSF-1 [33]. Within the TME, TAMs can quickly shift phenotypes in response to environmental signals. Generally, TAMs adopt an M1-like phenotype predominantly in the early stages of tumor development to eliminate abnormal cells and support a pro-inflammatory environment. Nonetheless, later phases are dominated by M2-like TAMs, which exhibit pro-tumoral and anti-inflammatory features [34]; in fact, M2-like TAMs exert immunosuppressive functions by releasing an array of anti-inflammatory and pro-tumoral cytokines such as Arginase 1 (Arg-1), transforming growth factor β (TGF-β), IL-6, and IL-10, and contribute to tumor spreading and angiogenesis, thereby supporting tumor progression [35,36]. The CSF-1R serves as a vital mediator for CSF-1 signaling, essential for governing the survival, differentiation, and proliferation of monocytes and macrophages [1]. In the context of the TME, CSF-1R activation is of primary importance. CSF-1 can be produced by tumoral cells and binds to the CSF-1R expressed on TAMs’ membranes. The subsequent dimerization and activation of the CSF-1R plays a critical role in recruiting and polarizing macrophages toward the M2 phenotype, due to the activation of downstream signaling pathways such as the phosphatidylinositol 3 kinase (PI3K)/AKT/forkhead box protein O1 (FOXO1) pathway, favoring tumor progression [37,38]. Additionally, CSF-1/CSF-1R interaction leads to the activation of PLC-γ2, STAT3, and ERK1/2 pathways, which culminates in the translocation of the transcription factor Sp1 into the nucleus, which in turn drives the differentiation of macrophages into the M2 phenotype, rendering the CSF-1R an important mediator of this cell population’s ability to dynamically shift from one phenotype to another [39]. The M2-like polarization of TAMs was evaluated by Van Overmeire and colleagues, who investigated the role of the CSF-1R in TAMs’ behavior and phenotype, especially in relation to their recruitment, extravasation, proliferation, and maturation. CSF-1R inhibition prevented the differentiation of Ly6C^hi^ monocytes into M2-like TAMs with a low expression of major histocompatibility complex (MHC)-II, therefore implying a direct effect of the CSF-1R in driving their polarization towards the M2-like phenotype [40]. In this context, a study that included co-culturing a macrophage cell line (U937) and an endometrial cancer (EC) cell line found that iNOS and CD86 expressions in U937 cells decreased, whereas Arg-1 and CD206 expression increased, indicating a shift from M1-like polarization toward a pro-tumoral M2-like polarization. In the same study, blocking the CSF-1R in U937 cells with PLX3397-inhibited EC cell proliferation by downregulating the expression of proliferation-associated proteins such as Janus kinase-1 (JAK1), PI3K, AKT, cyclin kinase 2, 4, and retinoblastoma-associated proteins [41]. Another study showed that IL-34 plays a pivotal role in regulating TAM polarization in colorectal cancer (CRC) through its interaction with the CSF-1R. IL-34, abundantly produced by CRC cells, acted as a cytokine that stimulated TAMs to adopt an M2-like phenotype, characterized by the increased expression of CD163 and CD206. Furthermore, IL-34 induced TAMs to produce IL-6, a cytokine known to enhance CRC cell growth and survival. The expression of IL-34 and the CSFR-1 by both neoplastic and non-tumoral cells within CRC tissue underscores its role in mediating crosstalk between cancer cells and the TME. Furthermore, this work proved that IL-34 was also produced by TAMs, suggesting a possible autocrine activation of IL-34/CSF-1R signaling in these cells [42]. In the context of AML, in a recent study using a mouse model induced by MLL-AF9, IL-34 overexpression accelerated disease progression and reduced survival [43]. Gene expression analysis revealed that IL-34 overexpression was associated with the upregulation of SOX13, a gene correlated with increased proliferation and LSC levels. Furthermore, IL-34 significantly influenced the leukemia microenvironment by increasing the number of TAMs which exhibited an M2-like phenotype. These M2-like TAMs expressed high levels of M2-associated genes such as CD206, ARG-1, IL-10, and MMP9, and displayed reduced phagocytic potential. This phenotype contributed to the pro-leukemic environment, facilitating AML progression [43]. A very recent study showed that the CSF-1R not only mediates TAMs’ polarization but also their recruitment to tumors. The authors developed an in vitro vascularized tumor model to study TAM recruitment and the impact of immunotherapy based on a multispecific CSF-1R/CCR2/TGF-β antibody in breast and lung cancers, and their results demonstrated that the CSF-1R is crucial for recruiting monocytes to the tumoral area while also mediating their pro-tumorigenic M2-like phenotype shift [44]. The clear implication of the CSF-1R in modulating pro-tumorigenic M2-like polarization of TAMs has shone a light on the therapeutic use of CSF-1R targeting to repolarize TAMs towards anti-tumorigenic M1-like phenotypes. For example, a study discovered orally active and selective inhibitors of the CSF-1R through the optimization of the clinical multitargeting kinase inhibitor BPR1K871. Molecular docking revealed a unique interaction between the inhibitor’s scaffold and the CSF-1R hinge region, enhancing its potency. The selected compound BPR1R024 exhibited potent in vitro CSF-1R inhibition and specifically targeted pro-tumorigenic M2-like TAMs while sparing anti-tumor M1-like TAMs. In vivo studies showed that the oral administration of BPR1R024 delayed tumor growth and shifted the immune landscape of the TME by increasing the ratio of M1-like and M2-like TAMs [45]. Similarly, an interesting study achieved the dual blockade of CD47/SIRPα and CSF-1/CSF-1R axes using a supramolecular assembly. This assembly was crafted from the small molecule AK750, known for interfering with the CSF-1R signaling pathway, and incorporated SIRPα antibodies to achieve the specific targeting of TAMs. In both B16/F10 melanoma and 4T1 breast cancer models, the supramolecule demonstrated robust anti-tumor and anti-metastatic efficacy, suggesting that the simultaneous blockade of the CD47/SIRPα and CSF-1/CSF-1R signaling axes can be further explored as a potent immunotherapeutic strategy [46]. Notably, an interesting study developed a sophisticated drug delivery system to administer CSF-1R siRNA, using an amphiphilic cationic cyclodextrin and DSPE-PEG-M2pep, a ligand that specifically targets M2-like TAMs, whose application was confirmed in vivo and in vitro. In addition, the nanoparticles not only effectively reprogrammed M2-like TAMs but also reshaped the TME, thereby facilitating the infiltration of CD4^+^ and CD8^+^ T cells while diminishing the immunosuppressive T_reg_ population [47]. These results strongly support the therapeutic efficacy of M2-like TAM repolarization mediated by CSF-1R inhibition [47]. Another study developed a lipid-based nanoparticle system capable of co-delivering CSF-1R and MAPK inhibitors to repolarize TAMs towards the M1-like phenotype. Their dual-inhibitor-loaded supramolecular nanoparticle system exhibited the sustained inhibition of CSF-1R and MAPK signaling pathways, leading to TAM repolarization over time in both in vivo and in vitro models, while also allowing for an increased infiltration of CD8^+^ T cells in the TME, enhancing the anti-tumor effect of TAM repolarization [48]. In this context, other groups confirmed that CSF-1R blockade in TAMs can induce the development of an anti-tumor and immunostimulatory milieu within the TME. For example, a study showed that the early and sustained depletion of TAMs using an anti-mouse CSF-1R antagonist antibody robustly inhibited tumor growth via T cell immune-mediated mechanisms in BALB/c mice models bearing mouse renal adenocarcinoma. This effect was associated with an elevated CD8^+^ T cell-to-T_reg_ ratio and the upregulation of genes associated with T cell recruitment and effector responses, even though these responses were tumor model-dependent [49]. Another study in sarcomas highlights the importance of targeting and reprogramming TAMs in the TME by investigating the impact of PLX3397, a potent CSF-1R inhibitor. Through cytokine array analysis and in vitro experiments using tumor-conditioned media, the authors not only demonstrated the role of CSF-1 in TAM activation and migration but showed that PLX3397-mediated CSF-1R inhibition effectively suppressed ERK1/2 phosphorylation, M2-like polarization, and chemotaxis. Performing in vivo studies in an osteosarcoma model employing PLX3397, they also revealed a significant inhibition of primary tumor growth and lung metastasis, accompanied by M2-like TAM and T_reg_ depletion, while also inducing the enhanced infiltration of CD8^+^ T cells within the TME [23]. Another study in CRC showed that blocking the CSF-1R with PLX3397 in multiple murine models reduces M2-like TAM infiltration and enhances CD8^+^ T cell infiltration, validating again the role of this receptor in shaping the immune landscape of the TME [50], and the same was demonstrated in very recent studies [51,52]. In line with these findings, another study showed that CSF-1R blockade in advanced prostate cancer, coupled with therapeutically restoring dysfunctional NOS3 levels, resulted in the increased infiltration of immunostimulatory cellular populations within the TME, again consisting of anti-tumoral M1-like TAMs, cytotoxic CD8^+^ T cells, B cells, and effector CD44^+^ CD62L T cells [53]. The same was demonstrated using a mouse model of Sonic Hedgehog medulloblastoma, where CSF-1R blockade using PLX5622 led to a clear shift in the intratumoral immune cell dynamics, confirming that the inhibition of the CSF-1R not only reduces pro-tumorigenic M2-like TAM populations but also facilitates the infiltration of CD8^+^ cytotoxic T cells into the TME [54]. Another study, which used CRISPR-Cas9 CSF-1 knockout murine breast (4T1) and colon (MC38) carcinoma models, showed that the inhibition of CSF-1R signaling boosts an immune-permissive TME, and results in improved tumor control compared to mice with parental tumors. In particular, in the MC38 CSF-1^−/−^ model, there was a notable upregulation of inflammatory chemokines like CCL20 and CCR9, associated with DC recruitment and the infiltration of CD8^+^ T cells into tumors. Consistently, the 4T1 CSF-1^−/−^ model exhibited a high expression of CCL2, CCL7, and CCL5, known to facilitate the migration and activation of immature DCs, and the expansion of MDSCs was attenuated [55]. A recent study performed using a myeloid-lineage-specific ubiquitin-specific peptidase 18 (USP18)-deletion mouse model additionally shows the importance of CSF-1R regulation within the TME, which was linked to its degradation and downregulation. Specifically, this group showed that USP18, a negative regulator of type I interferon, inhibited CSF-1R ubiquitination and degradation by interfering with the interaction between the CSF-1R and neural precursor cell-expressed developmentally downregulated 4 (NEDD4), an E3 ubiquitin ligase of the CSF-1R, also demonstrating the influence of USP18 on ubiquitin-conjugating enzyme H5 (UBCH5) expression, another protein implicated in CSF-1R degradation. USP18 deletion in myeloid cells slowed tumor growth, enhanced CD8^+^ T cell activity, and increased M1-like TAMs. In addition, USP18 deletion boosted the interaction between NEDD4 and the CSF-1R and increased UBCH5 expression, leading to enhanced CSF-1R degradation via the proteasome, thereby validating the importance of this receptor in driving the immune milieu of the TME [56]. The therapeutic inhibition of the CSF-1R to mediate TAM depletion was evaluated in a very recent study where a photodynamic self-assembly drug composed of CSF-1R inhibitor BLZ945 and chlorin e6 was used to deplete TAMs, reversing the immunosuppressive TME and enhancing the efficacy of photodynamic immunotherapy, which overcomes the nonimmunogenic tumor phenotype and promotes systemic anti-tumor responses [57]. The importance of the CSF-1R in TAMs is not limited to its role in modulating polarization within the TME as in fact other CSF-1R functions have been described. For example, TAMs’ ability to support tumor invasion has been thoroughly investigated, and it relies on the CSF-1R-mediated activation of PI3K and Src family kinase (SFK) pathways [58]. Another study reported that CSF-1R-ERK5 signaling supports TAM proliferation by suppressing p21 expression [59], and CSF-1R-induced local macrophage proliferation is a common hallmark of human tumors and a potentially important prognostic marker of malignancy [60]. In addition, it has been reported that IL-34, produced by hepatocellular carcinoma (HCC) cells, recruits TAMs to tumor sites, and correlates with metastasis and poor prognosis in HCC. IL-34 promotes TAM recruitment and viability via CSF-1R signaling. TAMs, in turn, enhance tumor growth, invasion, and metastasis through a paracrine loop with cancer cells. This process relies on the activation of the FAK and ERK1/2 signaling pathways. Additionally, high IL-34 levels are linked to increased immunosuppressive T_regs_ and decreased cytotoxic T cells within the tumor, aiding tumor immune evasion and progression [61]. Moreover, IL-34 exerted a profound influence on TAMs via CSF-1R signaling in ovarian cancer as in fact IL-34 facilitated the differentiation of monocytes into TAMs expressing membrane IL-1α (mIL-1α) and IL-18. These immunoregulatory macrophages, induced by IL-34, play a critical role in promoting a pro-inflammatory milieu within the TME. Specifically, they stimulate the transformation of non-Th17-committed memory CD4^+^ T cells into Th17 cells, characterized by the expression of CCR4, CCR6, and CD161, along with IL-17 and interferon-gamma (IFN-*γ*) production. This process is dependent on mIL-1α, which is constitutively expressed by TAMs and facilitates the Th17 cell differentiation pathway. This mechanism underscores IL-34 as a pivotal mediator in the intricate interplay between TAMs and T cells, thereby contributing to the chronic inflammation associated with angiogenesis and metastasis in ovarian cancer [62]. A study conducted to assess the prognostic significance of CSF-1 and CSF-1R expression in the tumor epithelial and stromal compartments of primary breast cancer and metastatic lesions in axillary lymph nodes, along with CD68^+^ TAMs and CD3^+^ T cells, revealed high expressions of these markers and high TAM and CD3^+^ T cell density in metastatic breast cancer compared with primary tumors, indicating a potential relationship with tumor aggressiveness [63]. Thus, the co-expression of CSF-1 and the CSF-1R in cancer cells suggests both autocrine and paracrine mechanisms driving tumor invasion and metastasis as the CSF-1R is highly expressed by TAMs, emphasizing the prognostic value of TAMs, and in turn of the CSF-1R, in breast cancer [63]. The prognostic value of CSF-1R expression in TAMs was also evaluated in lung adenocarcinoma, where it was examined for its association with mortality rates and patient smoking status [64]. In this work, the authors used a quantitative phosphor-integrated dot staining of the CSF-1R on CD68^+^ CD163^+^ TAMs in 195 lung adenocarcinoma cases and discovered that the presence of CSF-1R-expressing TAMs is associated with high mortality rates particularly in never-smoking patients, implying that CSF-1R-expressing TAMs might wield more potent tumor-promoting effects in never-smoking lung adenocarcinoma patients, making the CSF-1R a possible therapeutic target in this specific cohort of patients [64]. Other studies focused on evaluating resistance to treatment with CSF-1R inhibitors or exploiting CSF-1R inhibition to overcome chemotherapy resistance. A recent study proposed a different strategy to overcome treatment resistance in CRC, exploiting CSF-1R inhibition to reduce the infiltration of immunosuppressive TAMs [65]. The authors, using in vitro and in vivo studies, demonstrated that CRC cells become resistant to chemotherapy due to the upregulation of PD-L1 triggered by TGF-β. Chemorefractory cancer cells recruit TAMs via the CSF-1R, further enhancing PD-L1 expression through TGF-β signaling, thereby creating an immunosuppressive TME resistant to chemotherapy [65]. Upon the dual inhibition of CSF-1R and TGF-β signaling, this group showed that the proportion of immunosuppressive TAMs decreased, leading to the enhanced efficacy of chemotherapy and the reshaping of the TME, while also increasing the influx of cytotoxic CD8^+^ T cells and effector memory CD8^+^ cells, ultimately overcoming acquired chemoresistance in CRC [65]. Finally, an interesting study performed on mice delved into the intriguing process of B cell-to-macrophage transdifferentiation, as it revealed that B cell precursors, particularly CSF-1R^+^Pax5^Low^ pre-B and immature IgM^+^ B cells (BMBPs), possess remarkable plasticity, enabling them to differentiate into macrophages, which was proven as a mechanism exploited by cancer cells to generate immunosuppressive TAMs. The key orchestrator in this mechanism is the CSF-1R, which plays a central role in facilitating this transdifferentiation process. Specifically, CSF-1R signaling primes BMBPs for macrophage differentiation by inducing changes in chromatin accessibility, making this cell population more receptive to macrophage differentiation signals [66]. BMBP-derived macrophages (B-MFs) exhibited unique gene expression profiles and metabolic functions compared to macrophages derived from monocytes. Additionally, B-MFs contributed significantly to tumor progression by adopting immunosuppressive phenotypes and promoting cancer growth through the suppression of anti-tumor IFNγ^+^ CD4^+^ T cells and the induction of FoxP3^+^ T_regs_. Moreover, the origins of TAMs are diverse, with TAMs originating from both monocytes and BMBPs. However, the pivotal role of the CSF-1R in facilitating BMBP transdifferentiation into B-MFs highlights its potential as a target for therapeutic intervention aimed at disrupting the generation of immunosuppressive TAMs and enhancing anti-tumor immune responses. Interestingly, this mechanism is not only restricted to mice as B-MF-like cells and the related transcriptional signature can be observed in cohorts of metastatic or recurrent triple-negative breast cancer and high-grade serous ovarian cancer patients [66].

#### CSF-1R in the Microglia Associated with the TME of CNS Malignancies

Microglia constitute the major innate immune defense of the CNS, acting as tissue-resident macrophages [67]. Among other stimuli, the CSF-1/CSF-1R axis is pivotal in regulating their survival and proliferation within the CNS [68,69]. In addition, CSF-1R signaling regulates microglia development from yolk sac microglial progenitor cells as well as their homeostasis [70,71]. The importance of the CSF-1R in microglia has been further demonstrated by experiments showing that inhibiting the CSF-1R leads to a significant reduction in microglial cell density in the CNS [70,72]. This regulatory mechanism extends beyond normal CNS function and plays a critical role in the TME of CNS malignancies. The involvement of the CSF-1R in CNS tumors, including glioma and glioblastoma, has been extensively reviewed [73,74]. In the TME, the pool of microglia and TAMs recruited from circulating monocytes attracted by tumor-derived chemokines [75], known as glioma-associated microglia/macrophages (GAMMs), constitutes a significant portion of the glioma and glioblastoma mass. These cells support tumor progression through various mechanisms, including promoting immune evasion, angiogenesis, and tumor cell invasion [76]. GAMMs adopt a pro-tumorigenic phenotype influenced by tumor-derived cytokines, including CSF-1, which correlates with increased tumorigenesis and poor prognosis [76,77]. Targeting the CSF-1/CSF-1R axis has shown potential in reducing GAMM density and altering their phenotype towards a less tumor-supportive state [78,79,80]. In preclinical models, CSF-1R inhibition led to decreased tumor growth and invasion [78,81], although therapeutic efficacy varied depending on the glioma subtype [82], possibly depending on the heterogeneity of GAMM populations discovered through a recent single-cell RNA sequencing study [83].

Nevertheless, a recent study investigated the possibility of reprogramming GAMMs to improve glioblastoma therapies by targeting the CSF-1R using three CSF-1R inhibitors, PLX3397, BLZ945, and GW2580, on GAMMs isolated from patient tumors. GW2580 was the most effective, transforming immunosuppressive GAMMs into a proinflammatory phenotype, which was associated with the downregulation of M2 markers, the upregulation of M1 markers, the enhancement of MHC-II presentation, phagocytosis, and T cell-mediated tumor cell killing. Consequently, tumor cell proliferation in patient-derived organoids was reduced [84]. In contrast, PLX3397 was ineffective, aligning with its clinical trial failures [84]. Despite promising preclinical results, CSF-1R inhibitors have shown limited efficacy in clinical trials for glioblastoma [85] as in fact resistance to CSF-1R blockade is often observed, driven by compensatory mechanisms within the TME, including increased insulin-like growth factor-1 (IGF-1) signaling. Using genetic mouse models of glioblastoma multiforme (GBM), researchers demonstrated that while the overall survival significantly increased in response to CSF-1R inhibition, tumors eventually recurred in over 50% of mice. Nonetheless, when recurrent tumor cells were isolated and transplanted into naive animals, they observed that gliomas regained sensitivity to CSF-1R inhibition, suggesting that resistance to CSF-1R inhibition is driven by the TME. The RNA-sequencing analysis of both tumoral cells and GAMMs revealed elevated PI3K pathway activity in recurrent GBM, driven by GAMM-derived IGF-1 and tumor cell IGF-1 receptor (IGF-1R) [86]. Another study found that another possible resistance mechanism may occur by the recruitment of immunosuppressive lymphocytes upon CSF-1R blockade [87]. Furthermore, the CSF-1R inhibitor pexidartinib demonstrated poor CNS penetration, adding an additional layer of complexity to the potential use of this therapy [88]. Additionally, microglia are significantly affected by CSF-1R inhibition, with yet uncertain impacts on postnatal development. In pediatric patients, this may lead to potential adverse effects during microglia repopulation after drug removal [89]. Despite these challenges, combining CSF-1R inhibitors with other treatments, such as ionizing radiation (IR), has shown enhanced therapeutic efficacy. Combining IR with CSF-1R inhibitors prevented the recruitment and differentiation of pro-tumorigenic GAMMs, enhancing the efficacy of radiotherapy, thereby significantly extending survival in mouse models [90]. Another study found that the selective targeting of GAMMs was insufficient to enhance survival, but combining IR with continuous CSF-1R inhibition reverted IR-associated GAMMs, while extending survival and delaying disease relapse [91].

### 2.2. Myeloid-Derived Suppressor Cells

Because MDSCs act as one of the main actors in inducing immunosuppression and resistance to immunotherapy [92], deciphering the role of MDSCs is therefore crucial to understand the complexities of the TME. MDSCs are considered tumor-promoting cells, and their abundance in tumors correlates with worse prognosis [93]. In mice, MDSCs can be categorized into two different populations: polymorphonuclear (PMN)-MDSCs (CD11b^+^Ly-6G^+^Ly-6C^int/low^) which show typical neutrophilic characteristics, and monocytic (MO)-MDSCs (CD11b^+^Ly-6G^-^Ly-6C^hi^), similar to monocytes [94]. At the moment, cells identified as CD33^+^HLADR^low/neg^ are recognized as MDSCs. Concurrently, there are different subpopulations of MDSCs, characterized by distinct expression markers [95]. In both humans and mice, cytokines, including M-CSF, G-CSF (CSF-3), and GM-CSF, play a central role in the process of myeloid cell evolution [95,96]. Although the fundamental involvement of GM-CSF in the generation and mobilization of MO-MDSCs and G-MDSCs from the bone marrow is widely recognized [97], G-CSF is implicated in the accumulation and initiation of immunosuppressive action in PMN-MDSCs [98]. At the same time, CSF-1 and CSF-1R signaling are associated with the induction of immunosuppressive functions in MO-MDSCs [99]. As MO-MDSCs migrate to the tumor site, guided by various cytokines, they can undergo differentiation into TAMs. The precise mechanisms governing this transformation remain unclear, but potentially implicate cell mediators like hypoxia-inducible factor (HIF-α) and an abundance of pro-inflammatory cytokines [100]. The expression of the CSF-1R was observed in mouse MDSCs [101]. Nonetheless, although high CSF-1R levels were detected in specific tumor models, its expression in MDSCs remained inconsistent across different tumor types [102]. Furthermore, CSF-1R expression was used by Zou et al. to characterize MDSCs, considering the existence of both CSF-1R^+^ MO-MDSCs and CSF-1R^-^ MO-MDSCs. From this study, it is evident that CSF-1R^-^ MO-MDSCs show gene expression patterns similar to neutrophil granulocytes with a high expression of Olfactomedin 4 (OLFM4), thereby having a strong potential to transform into PMN-MDSCs, compared to CSF-1R^+^ MO-MDSCs.

CSF-1R^-^ OLFM4^hi^ MO-MDSCs are predominantly localized in peripheral tissues such as blood and bone marrow [103,104], whereas they are less abundant in the tumor mass. Conversely, CSF-1R^+^ MO-MDSCs do not show the expression of genes related to neutrophil cells. However, both CSF-1R^+^ and CSF-1R^-^ MO-MDSCs can differentiate into M2-like TAMs, indicating that there are likely different polarization mechanisms which could be independent of CSF-1 [104]. Various papers reported a link between the CSF-1R and MDSCs [101,105,106,107], but CSF-1R expression has only been detected in MO-MDSCs, and not in PMN-MDSCs [108,109]. Studying the cellular expression of the CSF-1R in patients with AML, Edwards et al. noticed how CSF-1R^hi^ cells in AML patients more frequently show the co-expression of myeloid-specific markers, compared with cells from healthy donors. Specifically, they show that these cells co-express HLA-DR and CD33, which are associated with classically activated macrophages and MDSCs, suggesting that CSF-1R^hi^ cells in AML may display a phenotype resembling MDSCs [110]. Other studies investigated CSF-1R targeting as a means to ablate MDSCs by reducing their pro-tumor activity. Holmgaard et al. used PLX647 to inhibit the CSF-1R in B16 murine melanoma cell lines transfected with indoleamine 2,3-dioxygenase (IDO), a dioxygenase which contributes to a worse prognosis by inducing the recruitment of MDSCs to the tumor site [111], and detected a reduction in CD11b^+^ Gr1^int^ MDSCs with a subsequent repolarization towards anti-tumoral behavior and an increase in cytotoxic lymphocyte tumor infiltration and activity [92]. Xu et al. further demonstrated that PLX3397 and GW2580 reduced tumor-infiltrating myeloid cells, including MDSCs, and tumor growth after treatment with irradiation in RM-1 prostate tumor-bearing mice. Interestingly, they also noted that using PLX3397 reduced the number of MO-MDSCs and PMN-MDSCs [112], whereas it was previously demonstrated that GW2580 only significantly decreased the number of MO-MDSCs [113]. Priceman et al. also showed that blocking the CSF-1R with GW2580 in tumor-bearing mice resulted in a significant reduction in the number of myeloid cells in the bone marrow and spleen, as well as decreased MO-MDSCs migrating from the blood to tumors [113]. Zhu et al. evaluated the gene expression profile in pancreatic ductal adenocarcinoma samples after treatment with PLX3397, observing a reduction in tumor-infiltrating MO-MDSCs but not in PMN-MDSCs and the downregulation of genes related to monocytic infiltration and pathways such as inflammatory response, chemotaxis, and myeloid leukotic-mediated immunity. This suggests how the CSF-1R is involved in immunosuppression and the blockade of the T cell-mediated immune response [22]. In a murine model, PLX3397 administration significantly increased the number of antigen-presenting macrophages and CD8^+^ T cells with a simultaneous reduction in MDSCs within tumor tissues, indicating an anti-tumoral shift in the TME [114]. Holmgaard et al. studied the use of CSF-1R inhibitory antibody (CS7) to evaluate the impact of blocking the receptor on the myeloid component of the breast TME. Although there was no significant overall reduction in CD11b^+^ myeloid cells after treatment, the authors discovered a significant selective impairment of MO-MDSCs when compared with PMN-MDSCs in the presence of CSF-1R inhibitors. Interestingly, they also observed an increase in MHC-II expression and INFγ production, while there was a reduction in the expression of immunosuppressive molecules such as Arg-1 and TGF-β, supporting the hypothesis that CSF-1R targeting does not lead to the depletion of MO-MDSCs, but rather a reorientation of these cells toward an anti-tumor phenotype, thereby favoring T cell-mediated immunity. Importantly, the precise mechanisms of this shift are not fully understood, underscoring the need for further research to elucidate these processes at the molecular level [108]. Moreover, in a very recent study, Tong et al. reported that the siRNA suppression of the CSF-1R affects the MDSC population, impacting the JAK/STAT3 pathway [115].

Furthermore, IL-34 produced by cancer cells is likely involved in MDSC-mediated tumor progression (Figure 2). Particularly, Kajihara et al. examined the effect of IL-34 on MDSCs in the TME of triple-negative breast cancer (TNBC), observing an increase in MO-MDSC differentiation from bone marrow and a decrease in PMN-MDSC after IL-34 induction [24]. This response helped in generating a more immunosuppressive microenvironment which attenuates the T cell-mediated anti-tumor response and enhances tumor growth. In addition, the authors demonstrated that IL-34 decreased the differentiation of myeloid stem cells into PMN-MDSCs, thereby inducing chemoresistance [24]. In ductal pancreatic cancer, tumor circulating cells produce IL-34, which promotes the differentiation of myeloid progenitor cells into MDSCs. Immunosuppressive molecules produced by these cells include nitric oxide (NO), Arg-1, and reactive oxygen species (ROS), which induce T lymphocyte anergy, consequently promoting tumor survival and proliferation [116].

### 2.3. Tumor-Associated Dendritic Cells

DCs, known to have both lymphoid and myeloid origins, have been associated with CSF-1R action in several studies. The CSF-1/CSF-1R axis in these cells plays a role in modulating survival, development, and migration, as well as in the release of inflammatory mediators under inflammatory conditions as reviewed by Xiang et al. [12]. Moreover, several studies have shown that DCs are regulated by CSF-1 signaling, which is essential for growth and development [117,118]. An interesting study conducted on renal carcinoma cells demonstrated how tumor cells can influence the development of CD34^+^ progenitor cells into mature DCs through the production of IL-6 and CSF-1, which in turn bind to the CSF-1R. This functional interaction promotes the development of monocytic-like cells which lack the ability to function as antigen-presenting cells, although possess phagocytic capabilities. Consequently, this event prevents the recognition of tumor cells by the immune system [119]. Moreover, Becker et al. described how tumor-produced CSF-1 and IL-6 can contribute to switching DCs from an anti-tumor phenotype to DC3s, which induce weak tumor antigen-specific CD8^+^ T cell activation (Figure 2). They demonstrated that the inhibition of CSF-1R and IL-6 in melanoma and non-small cell lung cancer (NSCLC) prevented tumor-induced DCs switching and enhanced the T cell-mediated immune response [117]. Furthermore, the expression of the CSF-1R by DC precursors is well documented [16,120,121]. However, the literature does not provide a clear consensus on whether DCs express the CSF-1R receptor. Various studies suggest that the CSF-1R is likely expressed by a subset of DCs, potentially serving specific functions within this pool. As validated by Graber et al., the CSF-1R gene is expressed by DCs in the spleen and lymph node T cell areas, but only a subset of these cells expresses the CSF-1R on the cell surface. Additionally, it has been confirmed that not all the subpopulations of DCs express the CSF-1R at the same levels, with DC2 cells expressing it more prominently [16,19]. Likewise, MacDonald et al. found that the c-FMS gene, responsible for encoding the CSF-1R, is highly expressed in the myeloid progenitors of DCs compared to plasmacytoid ones, which come from lymphoid progenitors. Additionally, the expression of the CSF-1R gene is more pronounced in tissue-resident DCs than in peripheral blood DCs. Despite this disparity, both DC subsets exhibit low levels of the CSF-1R protein on their cell membrane, with myeloid DCs showing slightly higher expression than plasmacytoid ones. Using a CSF-1^−/−^ mouse model, the authors also showed a decrease in both DC pools, confirming the essential role of CSF-1 in DC generation and differentiation [122] (Figure 2). In conclusion, due to the ambiguity in the literature regarding the expression of the CSF-1R in DCs, further comprehensive studies are necessary to understand the role of the CSF-1/CSF-1R axis in these cells, particularly in the TME.

## 3. CSF-1R in Tumor-Associated Stromal Cells

In the complex milieu of the TME, in addition to immune cells, various other cell populations play crucial roles in promoting cancer progression [14]. Among them, cancer-associated fibroblasts (CAFs), endothelial cells (ECs), and cancer stem cells (CSCs) express the CSF-1R and exhibit responsiveness to its regulatory signals (Figure 2).

### 3.1. Cancer-Associated Fibroblasts

CAFs are essential cellular components of the TME, significantly contributing to tumor progression, metastasis, immune evasion, and resistance to therapy [123,124]. Within the TME, multiple cellular precursors, including tissue-resident fibroblasts and non-fibroblast-derived cells, are prompted to differentiate into CAFs by a range of molecular signals [125,126]. Recent discoveries have highlighted the substantial influence of CAFs and the CSF-1/CSF-1R axis on the TME, affecting both tumor progression and immune responses. In fact, several studies have reported that CAFs are major producers of CSF-1, thereby shaping the immunological landscape of the TME (Figure 2). Under physiological conditions, fibroblastic-reticular cells, expressed in lymph nodes, influence resident macrophage and monocyte behavior and phenotype by stimulating CSF-1R signaling activated by CSF-1 secretion [127]. However, in tumors, CSF-1R signaling promoted by CAFs-derived CSF-1 has different implications. Mace and colleagues demonstrated that pancreatic stellate cells (PSCs), a subset of CAFs found within the TME of pancreatic cancer, secrete high levels of CSF-1 in addition to other soluble factors such as IL-6 and VEGF, which promote the differentiation and chemotaxis of MDSCs, known to suppress anti-tumor immune responses [128]. Along with these findings, in CRC, CAFs induce the differentiation of monocytes into CD68^high^/CD163^high^ macrophages. This effect is independent of cancer cells, underlining the pivotal influence of CAFs in dictating macrophage phenotype shift within the TME. Using in vitro co-culture experiments and cytokine profiling, the authors demonstrated that CAFs were major producers of CSF-1 and other soluble factors such as CCL2, crucial for macrophage recruitment and M2-like polarization, further emphasizing the critical role of the CSF-1/CSF-1R axis [129]. On this note, in pancreatic ductal adenocarcinoma, CAFs stimulated TAMs’ M2-like polarization by producing CSF-1 [130]. Another group showed that CSF-1 secretion by CAFs with other cytokines stimulated TAMs to express SPP1, not only influencing CD8^+^ T cell function but also promoting primary CRC [131]. Taken together, these results suggest that CAFs are major producers of the CSF-1R ligand CSF-1, thereby determining the immunological features of the TME. However, there is also evidence that CAFs express the CSF-1R on their cell membrane, suggesting that the CSF-1R may also have a role in regulating their activity. For example, Kumar and colleagues tested fibroblasts isolated from naive mice and found that these cells express both the CSF-1R and CXCL1. Treatment with CSF-1 induced the upregulation of the CSF-1R and the significant reduction in CXCL1 expression, supporting the hypothesis that the CSF-1R might regulate cytokine production by CAFs, further proving that these cells influence the recruitment of monocytes and macrophages to tumor sites and modulate PMN-MDSC behavior in the TME [132]. Another study demonstrated that the activation of the CSF-1R in CAFs leads to the secretion of macrophage inflammatory protein 2 (MIP2) which, upon binding to CXCR2 receptors on macrophages, compromises their cytotoxicity and enhances their angiogenic properties. Furthermore, CSF-1R inhibition in CAFs led to a reduction in IL-10 production and an increase in IL-12 production in macrophages and enhanced the cytotoxicity of macrophages against tumor cells, indicating a shift towards an anti-tumor M1-like phenotype [133]. Another study performed on CRC showed that CAFs not only produce IL-34 but also express its receptors, the CSF-1R and receptor-type protein-tyrosine phosphatase ζ (PTP-ζ). IL-34 enhanced the expression of CAF markers such as α-SMA, vimentin, and FAP in normal intestinal fibroblasts and promoted their proliferation. Moreover, the knockdown of IL-34 in CAFs reduced the expression of these markers and their proliferative capacity and decreased the ability of these cells to induce the migration of cancer cells. In addition, IL-34-stimulated CAFs produced factors like netrin-1 and b-FGF, which are crucial for CRC cell growth and migration, highlighting that the CSF-1R, activated by IL-34, regulates pro-tumorigenic CAF action, contributing to CRC progression by enhancing tumor cell proliferation, migration, and the inflammatory environment within the tumor stroma [42]. Collectively, these findings underscore the critical role of CAFs and the CSF-1/CSF-1R axis in tumor progression and immune modulation within the TME and suggest potential therapeutic strategies aimed at enhancing anti-tumor immune responses.

### 3.2. Endothelial Cells

In the intricate landscape of the TME, ECs emerge as pivotal orchestrators of tumor progression and immune response. ECs facilitate the delivery of oxygen and nutrients to the tumor, fueling angiogenesis and favoring metastatic dissemination, underscoring their essential role in tumor initiation and expansion [134,135]. Moreover, the dynamic interplay between ECs and immune cells within the TME intricately shapes the tumor milieu, exerting both pro-tumorigenic and anti-tumorigenic effects [136]. Immune cell regulation within tumors relies on, among other stimuli, EC activation and remodeling, highlighting the symbiotic relationship between ECs and the immune system in dictating tumor fate [137]. Yet, the underlying molecular mechanisms orchestrating this intricate crosstalk remain poorly characterized, posing an obstacle to the development of effective immunotherapeutic strategies. Evidence from the literature indicates that ECs are a significant source of CSF-1, substantially contributing to the angiogenesis and immunosuppressive features of target cells, particularly macrophages. For example, a study focused on the interaction between ECs and hematopoietic progenitor cells, particularly on their differentiation and polarization. The authors established an in vitro co-culture system where bone marrow-derived hematopoietic cells interacted directly with liver sinusoidal ECs. Remarkably, upon co-culture, a subset of hematopoietic progenitor cells transmigrated across the endothelium, forming colonies beneath it. These colonies, predominantly composed of macrophages, exhibited active proliferation and displayed the characteristics of activated M2-like macrophages, marked by the upregulation of Tie2 and the expression of M2-associated markers such as Arg-1, CD206/Mrc1, and CD36. Notably, this work indicated that EC-induced colony formation relied on endothelial-derived CSF-1, which activated the CSF-1R expressed on their cell membrane. Furthermore, the study highlighted the in vitro and in vivo pro-angiogenic capabilities of these macrophages as in fact functional analyses revealed that they can accelerate angiogenesis, promote tumor growth, and form close associations with ECs, underscoring the significant role of the CSF-1/CSF-1R axis in angiogenesis [138]. In addition, EC-derived CSF-1 is essential for regulating monocyte pools in the bone marrow and spleen, in particular for the maintenance of Ly6C^−^ monocyte survival and differentiation and re-establishing monocyte homeostasis following inflammatory challenges [139]. Moreover, in neovascular age-related macular degeneration, under hypoxic conditions, HIF-1α upregulates the expression of CSF-1 in choroidal ECs [37] and CSF-1 expression is mediated by HIF-1α binding to the hypoxia response element in the CSF-1 promoter. This is strictly correlated with EC interaction with macrophages as in fact CSF-1 secreted by ECs promotes the M2 polarization of macrophages by activating the CSF-1R and the downstream activation of the PI3K/AKT/FOXO1 pathway. Consistently, M2-type macrophages enhance the proliferation, migration, and tube formation of ECs in a CSF-1/CSF-1R-dependent manner, thereby contributing to angiogenesis [37]. Similarly, a study investigated the role of CSF-1 produced by ECs in diabetic retinopathy and its impact on CSF-1R signaling in microglia. The interaction between ECs and microglia and the activation of the CSF-1R mediated by EC-derived CSF-1 were crucial as they orchestrated a cascade of inflammatory responses and vascular dysfunction [140]. Despite this, there is limited evidence suggesting that the same mechanism is conserved in cancer. The existing literature on this topic is insufficient, highlighting the need for further research to enhance our understanding of these regulatory mechanisms and the role of endothelial cells in producing CSF-1 in cancer. Similarly, CSF-1R expression in ECs is poorly studied in both physiology and diseases including cancer. Indeed, while the CSF-1R is predominantly associated with myeloid cells, its expression in ECs was demonstrated in a few studies. In particular, the CSF-1R was identified as a significant marker of endothelial progenitor cells (EPCs). Specifically, the CSF-1R is highly expressed in the CD16^+^ monocyte compartment of peripheral blood. This expression pattern suggests that the CSF-1R may play a crucial role in identifying an EPC subpopulation in the peripheral blood monocyte compartment [141]. In 2014, a study showed that capillary ECs in the mouse central nervous system express the CSF-1R, identifying these cells as novel IL-34 targets. This work demonstrated that IL-34 helped in maintaining blood–brain barrier (BBB) integrity by upregulating tight junction proteins, such as claudin-5 and occludin, which are crucial for BBB function. This protective mechanism against BBB disruption, often triggered by pro-inflammatory cytokines such as IL-1β and tumor necrosis factor α (TNF-α) or amyloid β, is mediated via the CSF-1R signaling cascade, with the cAMP-responsive element-binding protein (CREB) emerging as an important player [142]. In accordance with this study, more recent work indicated that the CSF-1R was also expressed in ECs in the brain and CSF-1R signaling in ECs influenced the expression and maintenance of tight junction proteins [143]. CSF-1R inhibition leads to the decreased expression of several tight junction proteins in ECs, thereby compromising BBB integrity. Accordingly, the study also showed that ECs with heterozygous CSF-1R expression exhibited decreased tight junction protein expression and increased permeability, highlighting the genotype-specific effects of CSF-1R signaling on EC function, hinting that even the partial loss of CSF-1R function can significantly impact EC behavior and BBB integrity [143]. In conclusion, while these findings highlight the significant role of the CSF-1R in ECs, particularly in maintaining BBB integrity and identifying EPCs, they represent only an initial understanding. There is still much to uncover, especially in the context of the TME where various factors and cellular interactions could activate currently unknown functions of ECs mediated by the CSF-1R.

## 4. Cancer Stem Cells

Over the past few years, CSCs have attracted considerable interest in cancer research because of their central role in tumor initiation, progression, and resistance to therapy. CSCs have been identified in various tumor types, both solid and liquid, representing a small fraction of cells within the tumor mass and sharing several characteristics with normal stem cells. The uniqueness of these cells is the ability to self-renew and their plasticity, which allows them to adapt and survive modifications in the TME and therapies, resulting in tumor recurrence and metastasis. Understanding the interactions between CSCs and various cells in the TME is therefore necessary to better understand the function of these cells and develop more effective and targeted therapeutic strategies against cancer [144,145]. In fact, the elimination of CSCs is considered indispensable for effective anti-cancer treatment [146]. Several studies reported the involvement of the CSF-1/CSF-1R axis in stemness maintenance (Figure 2).

CSF-1R expression was detected at low levels in hematopoietic stem cells, and its expression increased along with stem cells’ commitment to the myeloid population [1,147]. Furthermore, the CSF-1R is critical for the maintenance of the intestinal stem cell niche and for the differentiation of Paneth cells [148]. Indeed, although *Csf-1r* mRNA expression is undetectable in intestinal crypt cells, it is present in crypt-associated macrophages, which are essential for maintaining the integrity of the small intestine. Notably, the prolonged blockade of CSF-1R determines the depletion of crypt-associated macrophages, with a significant impact on the differentiation of Paneth cells and stem cells expressing Lgr5, which represents a typical stemness marker in crypts [148]. Interestingly, CSF-1R^+^ stem cells are associated with oncogenic signals. In fact, the presence of an AIF1^+^ CSF-1R^+^ mesenchymal stem cell subpopulation has been identified in the inflammatory microenvironment in the liver. These CSF-1R-expressing cells play a pro-inflammatory role and promote macrophage infiltration, thereby stimulating the onset and progression of liver cancer [149]. Work by Cioce et al. investigated the expression profile of a subpopulation of CSF-1R^+^ mesothelioma cells, reporting the enrichment of several pluripotency-associated markers, including NANOG, OCT4, SOX2, ENDOGLIN, c-MYC, and NOTCH1, and chemoresistance genes such as *ABCG2* which is correlated with Pemetrexed resistance [8]. The presence of CSF-1R-expressing stem-like cells was also confirmed by Shi et al. in single-cell RNA sequencing experiments performed on primary CRC samples vis a vis normal mucosa samples [150]. The CSF-1R also plays a crucial role in AML induced by monocytic leukemia zinc finger gene (*MOZ*) fusions. Indeed, MOZ fusion proteins trigger strong CSF-1R transcription, which is dependent on PU.1 transcription factors, and leads to the constitutive expression of the CSF-1R in leukemia stem cells, thereby inducing AML [146]. Other studies have shown a link between the CSF-1R and CSCs as in fact Bayik and Lathia extensively analyzed the pivotal interaction between CSCs and immune cells within the TME, demonstrating that CSCs play a critical role in driving the polarization of TAMs through the CSF-1-CSF-1R axis across various cancer types [144]. Fan et al. explored the role of SOX9 in maintaining stemness and modulating the TME in the progression of gastric adenocarcinoma and demonstrated that the CSF-1R, expressed by TAMs in the TME, is involved in maintaining an immunosuppressive microenvironment that reduces the T cell-mediated response and induces stemness. The link between SOX9, expressed in tumor cells, and the CSF-1R, expressed by TAMs and particularly by the M2-like subset, is mediated by LIF/LIFR signaling. In fact, the study showed that the combined inhibition of LIF/LIFR and the CSF-1R leads to a reduction in tumor cells expressing high levels of SOX9, with a significant decrease in tumor growth, M2 macrophage infiltration along with TAM-mediated immunosuppression, the enhancement of the T cell-mediated response, and a reduction in SOX9-mediated tumor stemness [151]. Liu et al. studied the intricate link between miR-34a and the CSF-1R in intestinal tumorigenesis, showing that the upregulation of the CSF-1R, resulting from the loss of miR-34a, enhanced intestinal stem niche activation and increased Wnt signaling. The Wnt/β-Catenin pathway is crucial for cell–cell and cell–extracellular matrix interactions between the stem niche, which is believed to represent the initial cell population of intestinal carcinogenesis. Indeed, the loss of miR-34a and the increased expression of the CSF-1R increases cancer cell stemness, supported by the increased expression of stem cell markers, including Lrg5 and OLFM4 [152]. Moreover, Awasthi et al. highlighted the crucial role of the CSF-1R in regulating and maintaining CSCs in hepatocarcinoma. They demonstrated that inhibiting CSF-1R activity with a novel bioisostere of pexidartinib not only suppressed tumor growth in vivo but also inhibited CSC sphere formation in culture. This underscores the significance of the CSF-1/CSF-1R axis in sustaining CSCs [153]. This mechanism of CSF-1/CSF-1R action in maintaining CSCs was also demonstrated in lung cancer, where Hung et al. showed that the A549 cell line has a stem cell subset with a strong oncogenic phenotype. In vivo, CSF1KD mice exhibited reduced osteolytic activity, likely induced by CSF-1 produced by tumor cells, as well as decreased CSCs and osteoclastogenesis, thereby blunting tumor progression [154]. Moreover, in a study using 3D lung cancer cell culture models, Pass et al. showed that the suppression of CSF-1R kinase activity leads to decreased clonogenicity and 3D growth in several lung cancer cell lines, with the decreased expression of stemness and EMT markers (CD44, OCT4, SOX2, NANOG, VIMENTIN, and MMP-9), and the decreased expression of a chemoresistance gene (*ABCG2*) [155]. In addition, IL-34 is involved in CSC signaling through the CSF-1R in the TME. A pivotal study by Raggi et al. demonstrated that intrahepatic cholangiocarcinoma CSCs can form in vitro spheres and produce factors, including IL-34, which induce CD14^+^ macrophages to adopt a TAM-like phenotype, stimulating macrophage infiltration, differentiation, and the acquisition of a CSC-associated macrophage phenotype. Furthermore, IL-34 contributes to the maintenance of the stem-like properties of CSCs [156].

## 5. Conclusions

The CSF-1R emerges as a central player in shaping the immune landscape of the TME and targeting this receptor presents promising avenues promoting anti-tumor immunity and potentially improving clinical outcomes in various cancer types. Although CSF-1R expression and function have been extensively studied in myeloid cells, the role of this receptor in other immune cells, as well as stromal cells within the TME, has not yet been comprehensively explored. In this review, we conducted an in-depth analysis of CSF-1R expression across these various cell types in the TME and highlighted the significant implications for cancer progression (Table 1).

The diverse functions of the CSF-1R in regulating TAMs within the TME are crucial for understanding cancer progression and developing effective therapeutic strategies. In addition, the CSF-1R signaling pathway plays a significant role in shaping the TME of CNS malignancies, through its effects on GAMMs. While CSF-1R inhibition has demonstrated potential in preclinical studies, clinical translation has been challenging due to therapy resistance and the complex dynamics of the TME. To date, several comprehensive review articles have been published on the use of CSF-1R inhibitors, antagonists, and monoclonal antibodies in cancer therapy [36,105,157,158,159,160,161]. Future therapeutic strategies should focus on combinatorial approaches to effectively target multifaceted interactions within the TME and improve treatment outcomes for CNS-associated malignancies.

However, while this review provides a comprehensive overview of the role of the CSF-1R in the TME and highlights its potential as a therapeutic target, there are several limitations that should be acknowledged. One significant limitation is the fact that very few studies have investigated the expression of the CSF-1R in specific cells within the TME, such as ECs, DCs, and Tregs. Furthermore, some of these studies present results that are not entirely conclusive, making it difficult to draw firm conclusions on the CSF-1R’s role in these cell types. The function of these cells can be critical in tumor progression and shaping the TME, but how the CSF-1R contributes to these particular cell pools remains unclear. Additionally, the precise molecular mechanisms by which the CSF-1R exerts its effects within the immune and stromal TME cells remain only partially elucidated. This gap in knowledge hampers the ability to develop highly specific and effective therapeutic interventions.

As reported in a recent review [3], another limitation in the study of CSF-1R expression and function in tumors includes the fact that it is expressed at low levels in cancer cells compared to the monocytes/macrophages that represent the majority of CSF-1R^+^ cells, thereby making it difficult to detect in some experimental settings. The development of more sensitive approaches including RNAscope or new specific anti-CSF-1R antibodies should be investigated in tumor tissue.

In the future, research efforts should focus on conducting more in-depth studies to elucidate the molecular signaling pathways activated by the CSF-1R in various TME components, including TAMs, GAMMs, MDSCs, CAFs, and CSCs. Investigating the crosstalk between these cells and their role in therapy resistance will be crucial for designing effective combinatorial treatment strategies. Moreover, future research should prioritize the development of novel biomarkers to predict patient responses to CSF-1R-targeted therapies and the exploration of synergistic strategies that combine CSF-1R inhibition with other immunotherapies or targeted therapies. This comprehensive approach could ultimately enhance the clinical translation of CSF-1R-targeted treatments and improve patient outcomes across a broader spectrum of cancers.

## Figures and Tables

**Figure 1 biomedicines-12-02381-f001:**
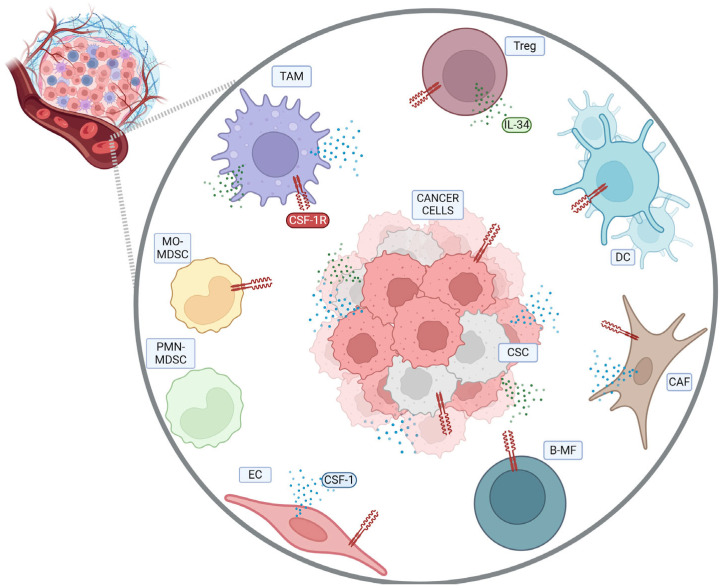
CSF-1R expression in tumor microenvironment cells. Within the TME, several cell types express the CSF-1R. These include cancer cells and CSCs. Among immune cells, CSF-1R expression is particularly noted in TAMs, Tregs, and DCs. Notably, within the DC population, only a subset of cells displays the CSF-1R on their cell surface. In the MDSC population, the CSF-1R is expressed exclusively by monocytic MDSCs (MO-MDSCs), and not by polymorphonuclear MDSCs (PMN-MDSCs). Additionally, CSF-1R expression in CSF-1R^+^Pax5^Low^ pre-B cells and immature IgM^+^ B cells (BMBPs) is crucial for the B cell-to-macrophage transdifferentiation process, a mechanism that cancer cells exploit for generating immunosuppressive TAMs. Among stromal cells, CAFs and ECs express the CSF-1R. The ligands for CSF-1R, CSF-1 (represented as blue dots in the figure) and IL-34 (represented as green dots in the figure), are produced by different cellular populations within the TME, with TAMs, CAFs, cancer cells, CSCs, and ECs producing CSF-1, and IL-34 being secreted by cancer cells, CSCs, TAMs, and Tregs. Created with https://www.biorender.com/ (accessed on 25 July 2024).

**Figure 2 biomedicines-12-02381-f002:**
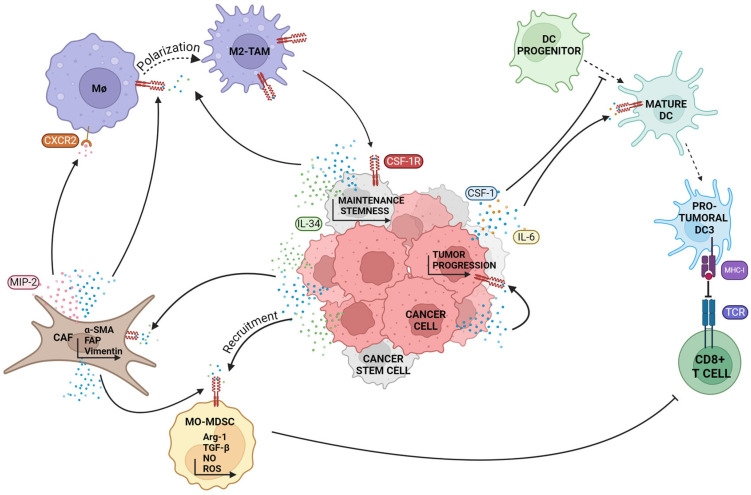
The CSF-1R axis mediates the crosstalk between tumor microenvironment cells. In the TME, the CSF-1/IL-34-CSF-1R signaling pathway enables cell communication, influencing phenotypes and cytokine production. CAFs secrete high levels of CSF-1, promoting the differentiation and chemotaxis of MDSCs and the recruitment of monocytes into M2-like macrophages. TAMs acquire M2-like phenotypes in response to CSF-1 and IL-34 from tumor cells. In CRC, IL-34 enhances CAF marker expression, particularly α-SMA, vimentin, and FAP. CSF-1R activation in CAFs leads to the secretion of MIP2, which affects macrophage function. In renal carcinoma, CSF-1 can impair the development of CD34^+^ progenitors into mature DCs. In melanoma and NSCLC, CSF-1 and IL-6 contribute to a switch to pro-tumor phenotype DCs (DC3s), reducing their antigen-presenting ability and impairing T lymphocyte activation. CSF-1 and IL-34 from cancer cells recruit and enhance the immunosuppressive functions of MO-MDSCs, inducing the production of NO, Arg-1, and ROS. The CSF-1/CSF-1R axis also mediates M2-like TAM polarization and maintains CSCs, with CSCs also producing IL-34, thus influencing the TME. Created with https://www.biorender.com/ (accessed on 5 July 2024).

**Table 1 biomedicines-12-02381-t001:** Role of CSF-1R in different components of the TME.

Cell Type	Functions	References
TAM	Promotes polarization to M2-like phenotype	[33,37,38,39,62]
	Facilitates recruitment to tumor sites	[33,44,61]
	Supports proliferation by suppressing p21 expression	[59]
	Supports tumor invasion via PI3K and Src family kinase pathways	[58]
GAMM	Regulates survival, proliferation, development from progenitors, and homeostasis within the CNS	[68,69,70,71]
	Promotes polarization towards pro-tumorigenic phenotype	[76,77]
MDSC	Promotes immunosuppressive functions	[22,92,99,108,114,116]
	Facilitates recruitment to tumor sites	[111,113]
DC	Induces survival, development, and migration	[12,117,118]
	Promotes polarization to tumor-associated phenotype	[117]
	Impairs antigen presentation	[119]
CAF	Promotes CAF marker expression, proliferation, and pro-tumoral molecules	[42]
	Regulates cytokine production, and reduces macrophage cytotoxicity and enhances their angiogenic properties	[132,133]
EC	Influences tight junction protein expression, contributing to BBB integrity	[143]
CSC	Involved in stemness maintenance	[144,145]
T_reg_	Unknown/To be investigated	[23,24,25]
MK	Unknown/To be investigated	[16,17,18]

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
