# Peer review of "Insights into CSF-1R Expression in the Tumor Microenvironment"

_biomedicines, 2024, doi:10.3390/biomedicines12102381_

Round 1
Reviewer 1 Report (Previous Reviewer 1)
Comments and Suggestions for Authors
The authors did a great job during the revision.
Author Response
We thank the reviewer for the positive comment.
Reviewer 2 Report (Previous Reviewer 2)
Comments and Suggestions for Authors
The authors need to address the following issues:
-
Include review tables in Sections 2 and 3.
-
Reorganize the manuscript into the following sections: Introduction, Literature Review, Materials and Methods, Discussion, and Conclusion.
-
Incorporate the ARIMA model to define the literature inclusion and exclusion criteria.
-
Use mathematical formulations or hypotheses to structure the review of the articles.
-
Provide a block diagram of the proposed methodology.
Author Response
Please see the attachment

Reviewer 3 Report (New Reviewer)
Comments and Suggestions for Authors
- The authors are advised to focus on the future of therapeutic applications and the clinical trials of CSF-1R.
- The authors need to discuss the challenges in detecting low CSF-1R expression in cancer cells and how they could be resolved.
- The authors need to add figures captions.
- The authors need to clarify the hypothesis in the introduction
- The references need more updated studies concerning the scope of the author.
- The authors need to discuss the dynamic nature of TAM plasticity.
- The authors need to discuss how CSF-1R affects CAF activation, tumor-stroma interactions, and immune cell trafficking
- The authors need to explain how CSF-1R-targeted therapies might overcome MDSC-mediated immunosuppression.
- The authors need to provide a comparative analysis of MDSC targeting strategies in different tumor types would be insightful.
- The authors are advised to discuss the mechanisms underlying CSCs' resistance to therapy.
- The authors need to highlight the challenges of CSF-1R inhibition
- The authors need to check some minor grammatical errors and awkward sentence constructions throughout the paper.
- The authors need to revise the inconsistency in terminology.
- All over the review, the consistency and flow need to be improved.
Minor English language is required
Round 2
Reviewer 2 Report (Previous Reviewer 2)
Comments and Suggestions for Authors
Still I am not stisfied from the authors response.
Reviewer 3 Report (New Reviewer)
Comments and Suggestions for Authors
The authors have responded to all comments appropriately
Comments on the Quality of English LanguageMinor editing of English language required.
This manuscript is a resubmission of an earlier submission. The following is a list of the peer review reports and author responses from that submission.
Round 1
Reviewer 1 Report
Comments and Suggestions for Authors
The Colony Stimulating Factor 1 Receptor (CSF-1R) has a crucial function in coordinating cellular interactions in the tumor microenvironment (TME). This review examines the diverse roles of CSF-1R in different cellular populations within the tumor microenvironment (TME), such as tumor-associated macrophages (TAMs), myeloid-derived suppressor cells (MDSCs), dendritic cells (DCs), cancer-associated fibroblasts (CAFs), endothelial cells (ECs), and cancer stem cells (CSCs). This process promotes tumor development and helps the tumor evade the immune system. The manuscript is well-articulated and holds immediate relevance for technicians in the health sciences. However, some constructive comments are provided to enhance the overall quality of the manuscript:
1) Although the authors have incorporated a substantial amount of literature, I kindly request that the limitations and future directions be elaborated upon in greater detail.
2) Please include the peculiarities of CSF-1R in shaping the immune landscape of the TME, promoting anti-tumor immunity, and potentially improving clinical outcomes in various cancer types.
3) Some more figures should be included to provide more specific details to the readers.
4) Researchers have neglected to consider some studies from recent investigations. Please incorporate multiple guidelines from the suggested articles within the study. For example: Decision Support System for the Prediction of Mine Fire Levels in Underground Coal Mining Using Machine Learning Approaches/ Application of KNN-based isometric mapping and fuzzy c-means algorithm to predict short-term rockburst risk in deep underground projects
5) The authors are requested to place greater emphasis on the novelty of the work.
6) Although this is a review article and the authors have provided literature based on their own selection, there is no discussion or conclusion section. Additionally, a methodology section is missing. Including these headings and elaborating on each section will make the paper more accessible and beneficial for readers.
While the study presents the therapeutic potential of targeting CSF-1R to reduce MDSC-mediated immunosuppression and enhanced anti-tumor immunity, addressing these comments will notably improve the quality of the proposed methodology.
Comments on the Quality of English LanguageMinor editing of English language required.
Reviewer 2 Report
Comments and Suggestions for Authors
The manuscript "Insights into the CSF-1R Expression in the Tumor Microenvironment" does not have sufficient contributions to be published in its current form.
1. No tables have been presented to list the reviewed literature.
2. The article appears to be a theoretical overview regarding CSF-1R expression.
3. The authors have not used the ARIMA model for literature inclusion and exclusion criteria.
4. No block diagram is presented to elaborate on the methodology utilized for the review.
5. The authors have not used mathematical formulations or hypotheses for reviewing the articles.
6. The structure of the article is not properly organized.
Comments on the Quality of English LanguageThere are many English grammatical, typographical, spelling, and sentence formation errors that need to be corrected.